



# Understanding the surface temperature response and its uncertainty to CO₂, CH₄, black carbon and sulfate

Kalle Nordling[1], Hannele Korhonen[1], Jouni Räisänen[2], Antti-Ilari Partanen [1], Bjørn H. Samset [3]  and  Joonas Merikanto[1]

[1]Finnish Meteorological Institute, Helsinki, Finland.

[2]INAR/Physics, University of Helsinki, Helsinki, Finland.

[3] CICERO Center of International Climate Research, Oslo, Norway.

*Correspondence to*: Kalle Nordling(kalle.nordling@fmi.fi)

**Abstract.** Understanding the regional surface temperature responses to different anthropogenic climate forcing agents, such

as greenhouse gases and aerosols, is crucial for understanding past and future regional climate changes. In modern climate models, the regional temperature responses vary greatly for all major forcing agents, but the causes of this variability are poorly understood. Here, we analyse how changes in atmospheric and oceanic energy fluxes due to perturbations in different anthropogenic climate forcing agents lead to changes in global and regional surface temperatures. We use climate model data on idealized perturbations in four major anthropogenic climate forcing agents ($CO_2$, $CH_4$, and sulfate and black carbon

aerosols) from PDRMIP climate experiments for six climate models (CanESM2, HadGEM2-ES, NCAR-CESM1-CAM4, NorESM1, MIROC-SPRINTARS, GISS-E2). Particularly, we decompose the regional energy budget contributions to the surface temperature responses due to changes in longwave and shortwave fluxes under clear-sky and cloudy conditions, surface albedo changes, and oceanic and atmospheric energy transport. We also analyse the regional model-to-model temperature response spread due to each of these components. The global surface temperature response stems from changes in longwave

emissivity for greenhouse gases ($CO_2$ and $CH_4$) and mainly from changes in shortwave clear-sky fluxes for aerosols (sulfate and black carbon). The global surface temperature response normalized by effective radiative forcing is nearly the same for all forcing agents (0.63, 0.54, 0.57, 0.61 $KW^{-1}m^2$). While the main physical processes driving global temperature responses vary between forcing agents, for all forcing agents the model-to-model spread in temperature responses is dominated by differences in modelled changes in longwave clear-sky emissivity. Furthermore, in polar regions for all forcing agents the

differences in surface albedo change is a key contributor to temperature responses and its spread. For black carbon the modelled differences in temperature response due to shortwave clear-sky radiation are also important in the Arctic. Regional model-to-model differences due to changes in shortwave and longwave cloud radiative effect strongly modulate each other. For aerosols clouds play a major role in the model spread of regional surface temperature responses. In regions with strong aerosol forcing the model-to-model differences arise from shortwave clear-sky responses and are strongly modulated by combined temperature

responses to oceanic and atmospheric heat transport in the models.



## 1 Introduction

Climate change projections depend highly on future scenarios of climate mitigation actions. But in addition to uncertainty arising from different possible futures particularly in timescales of decades the climate projection uncertainties are
dominated by the climate model response uncertainty (Hawkins & Sutton, 2009; Lehner et al., 2020). This arises from structural differences between different climate models. Climate models differ on how they represent the radiative forcing of anthropogenic greenhouse gases and aerosols. But perhaps more importantly, they respond differently to the same external radiative forcing (Nordling et al., 2019). As stated in Lehner (2020) the model spread in the estimated temperature responses is affected by inter-model differences in both the forcing and in how the models respond to the forcing.

Smith et al. (2020) quantified the effective radiative forcings (ERFs) for modern-day greenhouse gas and aerosol concentrations for a range of climate models participating to CMIP6 multi-model climate experiments. They showed that since CMIP5, the spread in modelled radiative forcing has narrowed down. Despite this, the response uncertainty in CMIP6 models appears to have grown from CMIP5 models (Lehner et al., 2020; Zelinka et al., 2020). Uncertainty in the climate response hampers efforts to robustly define carbon emission targets to maintain global warming below specified limits, such
as below 1.5 °C (Matthews et al., 2021; Rogelj et al., 2019). Furthermore, the carbon emission targets depend on the climate response to radiative forcers besides carbon dioxide, such as aerosols and methane (Tokarska et al., 2018). Modern day anthropogenic aerosols cool the global surface temperatures between 0.5-1.1 °C (Samset et al., 2018, Nordling et al., 2019), and their future reductions can accelerate global warming and enhance global precipitation (e.g. Merikanto et al., 2021; Wilcox et al., 2020).

Besides the need to better understand the impacts of different climate forcing agents on the global climate, there is an urgent need to better understand how they impact climate on regional scale. The spatial distribution of aerosols is highly heterogenous, and much of the modern-day effective aerosol radiative forcing is concentrated over the South and East Asian region (Fiedler et al., 2019), while the radiative forcing of long-lived greenhouse gases is much more uniform (Shindell et al., 2015). Aerosols have both local and remote climate effects which depend on the emission region and type of aerosol
(Merikanto et al., 2021; Nordling et al., 2019; Persad & Caldeira, 2018). Furthermore, the differences in regional distributions of aerosol surface temperature responses are not dominated by the aerosol description in modern climate models (Nordling et al., 2019). Therefore, differences in modelled regional temperature responses for both greenhouse gases and aerosols appear to mainly depend on differences in dynamic responses of the atmosphere-ocean-sea ice system in the models. The main focus of this paper are these differences in modelled responses to aerosol and greenhouse gas
perturbations in different climate models.

The Precipitation Driver Response Model Intercomparison Project (PDRMIP) (Myhre et al., 2017) provides a data set that allows us to investigate how different climate forcing agents affect the Earth's climate in global and regional scale. PDRMIP



comprised idealized single-forcer scenarios for several independent climate models. Previously, the PDRMIP data set has been used to study e.g. how different forcing agents affect the Arctic amplification (Stjern et al., 2019) and how they produce
rapid adjustments and ERF (Smith et al., 2018). Estimating ERF is not straightforward, and different methods provide a variety of different results. For example, Tang et al. (2019) used PRDMIP data to estimate ERF for different climate forcing agents with several different methods. The model-mean estimated ERF for the doubling of carbon dioxide concentrations varied from 3.65 to 4.70 Wm$^{-2}$ depending on the method and on how rapid adjustments were included in the estimate. Richardson et al. (2019) showed that ERF calculated from fixed-sea-surface experiments is a good predictor for the global
temperature change for different forcing agents, and particularly so if the adjustments due to land temperature change are included.

The model differences in climate response are often investigated through radiative feedback analysis (e.g. Zelinka et al., 2020). While the feedback analysis is particularly suitable for analyzing the root causes of model-to-model differences in the equilibrium climate sensitivity (the equilibrium temperature response to doubled atmospheric carbon dioxide concentrations),
it is less suitable for exploring regional temperature response variance between the models due to the nonlinearity of regional feedbacks (Andrews et al., 2012). Räisänen & Ylhäisi (2015) formulated an energy balance framework to explore the impact of the top-of-the atmosphere (TOA) radiative fluxes, atmospheric energy transport and the net surface energy flux on regional surface temperatures. The method relies on the local conservation of energy and it is therefore mathematically an almost exact solution for the decomposition of energetic components of the temperature response. Its also takes into account both the
horizontal energy transport and surface energy fluxes on the local energy balance. Räisänen (2017) included a more detailed shortwave radiative flux treatment according to Taylor et al. (2007), and Merikanto et al. (2021) included a cloud radiative kernel treatment for a more physical separation of longwave cloud and clear-sky radiative fluxes. In this paper, we use this energy balance framework with climate model data from PDRMIP experiments to study the origins of regional temperature response and its standard deviation in six different climate models to four different climate forcing agents (carbon dioxide,
methane, sulfate and black carbon). Evaluation of the mechanisms responsible for the model spread is key for understanding why models still exhibit a substantial spread in temperature response even when forced identically.

## 2 Materials and Methods

### 2.1 Decomposition of the surface temperature response







**Figure 1. Illustration of the local atmospheric energy budget in a single atmospheric column from the surface to the top-of model atmosphere (TOA). We attribute the change in local surface temperature to changes in different terms of the local energy budget. (a) Unperturbed conditions, where red arrows indicate longwave (thermal) radiation, yellow arrows indicate shortwave (solar) radiation, blue arrows indicate horizontal incoming and outgoing energy, and curvy red arrows indicate latent and sensible heat. Under perturbed conditions, we decompose the change in the energy budget to (b) the change in TOA solar radiation due to changes in surface albedo, and to the change in shortwave clear-sky flux (separately). The change in shortwave clear-sky flux is mainly caused by changes in aerosol concentrations or changes in atmospheric water vapor; (c) the change in longwave TOA flux, which is mainly caused by changes in atmospheric water vapor concentrations, atmospheric thermal structure (lapse rate) or greenhouse gas concentrations; (d) changes in longwave and shortwave TOA fluxes, separately, due to changes in cloudiness and cloud microphysics, (e) the combined change in surface energy balance, including the change in the net shortwave and longwave energy**





**flux into the surface and changes in latent and sensible heat fluxes; (f) the combined change in horizontal energy transport and internal energy of the atmospheric column, calculated from the convergence of energy.**

We attribute local surface air temperature response to different energetic components, namely to changes in local longwave clear-sky and cloud fluxes associated with change in emissivity ($\Delta LW^{\uparrow}_{TOA,clr,e}$ and $\Delta LW^{\uparrow}_{TOA,cld,e}$, respectively, with the arrow indicating the vector direction towards space) at TOA, shortwave clear-sky and cloud radiative fluxes ($\Delta SW^{\downarrow}_{TOA,clr}$ and $\Delta SW^{\downarrow}_{TOA,cld}$), changes in surface energy fluxes ($\Delta F^{\downarrow}_{SURF}$, essentially representing changes in atmosphere-to-ocean net heat flux), and convergence of atmospheric energy ($\Delta CONV$, representing horizontal atmospheric heat transport). These changes

are illustrated in Fig. 1.We use the method presented in Räisänen and Ylhäisi (2015), Räisänen (2017), and Merikanto et al. (2021). The method is based on a concept of planetary emissivity(Harshvardhan and Cess, 1976), which links the local surface air temperature ($T$) to the outgoing long wave radiation at the top of the atmospheric column ($LW^{\uparrow}_{TOA}$),

$$\varepsilon_{eff} = \frac{LW^{\uparrow}_{TOA}}{\sigma T^4}, \tag{1}$$

where $\varepsilon_{eff}$ is an effective local planetary emissivity and $\sigma$ is the Boltzmann constant. Then, letting [ ] to mark the mean state

between baseline and perturbed climates, the change in outgoing longwave radiation between the two climate states can be written as

$$\Delta LW^{\uparrow}_{TOA} = 4\sigma[\varepsilon_{eff}][T^3]\Delta T + \sigma\Delta\varepsilon_{eff}[T^4] = D\Delta T + \Delta LW^{\uparrow}_{TOA,\varepsilon} \tag{2}$$

where $D\Delta T$ is the local change in outgoing thermal radiation at constant emissivity (i.e at fixed thermal atmospheric structure and water vapor concentration), and hence $D$ represents the local Planck feedback parameter. $\Delta LW^{\uparrow}_{TOA,\varepsilon}$ is the change in the

outgoing thermal radiation associated with the change in the local emissivity of the atmosphere.

The rate of energy change within an atmospheric column is given by the energy balance equation

$$\frac{\delta E}{\delta t} = SW^{\downarrow}_{TOA} - LW^{\uparrow}_{TOA} - F^{\downarrow}_{SURF} + C^{\rightarrow}, \tag{3}$$

where $\frac{\delta E}{\delta t}$ is the change of internal energy within the column with respect to time, $SW^{\downarrow}_{TOA}$ is the net incoming flux of solar

radiation, and $C^{\rightarrow}$ is the horizontal transport of energy to the column, and the net downward heat flux $F^{\downarrow}_{SURF}$ into the surface is given by


$$F^{\downarrow}_{SURF} = SW^{\downarrow}_{SURF} + LW^{\downarrow}_{SURF} - SH^{\uparrow} - LH^{\uparrow}. \tag{4}$$

The change in $LW^{\uparrow}_{TOA}$ between two climate states can thus be written as

$$\Delta LW^{\uparrow}_{TOA} = \Delta SW^{\downarrow}_{TOA} - \Delta F^{\downarrow}_{SURF} + \Delta\left(C^{\rightarrow} - \frac{\delta E}{\delta t}\right) \tag{5}$$

Using Eq. (2) with Eq. (5), the local change in surface temperature can be decomposed to different energetic components as

$$\Delta T = -\frac{\Delta LW^{\uparrow}_{TOA,\varepsilon}}{D} + \frac{\Delta SW^{\downarrow}_{TOA}}{D} - \frac{\Delta F^{\downarrow}_{SURF}}{D} + \frac{\Delta\left(C^{\rightarrow} - \frac{\delta E}{\delta t}\right)}{D} = \Delta T_{LW} + \Delta T_{SW} + \Delta T_{SURF} + \Delta T_{CONV} \tag{6}$$

$\Delta T_{LW}$, $\Delta T_{SW}$ and $\Delta T_{SURF}$ can be calculated directly from the standard energy flux output of the models, and $\Delta T_{CONV}$ as a residual term. $\Delta T_{CONV}$ includes both horizontal energy transport and change in local atmospheric energy storage which is

insignificant at annual time scales. Furthermore, $\Delta T_{SW}$ can be decomposed into clear-sky, cloud, albedo and non-linear terms using the Approximate Partial Radiative Perturbation (APRP) method (Taylor et al., 2007),

$$\Delta SW^{\downarrow}_{TOA} = \Delta SW^{\downarrow}_{TOA,in} + \Delta SW^{\downarrow}_{TOA,clr} + \Delta SW^{\downarrow}_{TOA,cld} + \Delta SW^{\downarrow}_{TOA,albedo} + \Delta SW^{\downarrow}_{TOA,nl} \tag{7}$$

where $\Delta SW^{\downarrow}_{TOA,in}$ is the change in incoming solar radiation, $\Delta SW^{\downarrow}_{TOA,clr}$ is the change in net TOA solar radiation due to changes

in clear-sky radiative properties of the atmosphere, $\Delta SW^{\downarrow}_{TOA,cld}$ the change in net TOA solar radiation due to changes in clouds, $\Delta SW^{\downarrow}_{TOA,albedo}$ the change in net TOA solar radiation due to change in surface albedo, and $\Delta SW^{\downarrow}_{TOA,nl}$ is a non-linear correction term arising from the APRP method. $\Delta SW^{\downarrow}_{TOA,in}$ is constant if the incoming solar flux is constant. $\Delta SW^{\downarrow}_{TOA,nl}$ is typically negligibly small, and can be ignored (Räisänen, 2017; Merikanto et al., 2021).

Also $\Delta LW^{\uparrow}_{TOA,\varepsilon}$ can be decomposed into clear-sky (CS) and cloud radiative effect (CRE) components,


$$\Delta LW^{\uparrow}_{TOA,\varepsilon} = \Delta LW^{\uparrow}_{TOA,CS,\varepsilon} + \Delta LW^{\uparrow}_{TOA,CRE,\varepsilon}. \tag{8}$$

However, $\Delta LW^{\uparrow}_{TOA,CRE,\varepsilon}$ is affected by changes in non-cloud feedbacks (water vapor, surface albedo and air temperature), making it a negatively biased approximation of the actual cloud longwave feedback. To obtain a more accurate estimation of





the actual cloud contribution to longwave emissivity change, we applied the radiative kernel method of (Soden et al., 2008). With this method, a correction factor can be calculated,


$$\Delta LW^\uparrow_{corr} = \left(K_T - K_T^{clr}\right)\Delta T + \sum_i \left(K_{T_i} - K_{T_i}^{clr}\right)\Delta T_i + \sum_i \left(K_{w_i} - K_{w_i}^{clr}\right)\Delta(\ln q)_i, \quad (9)$$

where $K_T$, $K_{T_i}$ and $K_{w_i}$ represent radiative kernels where each state variable (surface temperature, temperature profile and water vapor respectively) is perturbed by unit change. The corrected clear-sky and cloud longwave emissivity changes then become

$$\Delta LW^\uparrow_{TOA,clr,\varepsilon} = \Delta LW^\uparrow_{TOA,CS,\varepsilon} + \Delta LW^\uparrow_{corr}, \quad (10a)$$


$$\Delta LW^\uparrow_{TOA,cld,\varepsilon} = \Delta LW^\uparrow_{TOA,CRE,\varepsilon} - \Delta LW^\uparrow_{corr} \quad (10b)$$

All results have been calculated using three different kernels (ECHAM (Block & Mauritsen, 2013), GFDL (Pendergrass et al., 2018) and HadGEM2 (Smith, 2018) to obtain a better estimate of the overall cloud effect. The correction factor of Eq. (9) has been calculated as an average of the three kernels.

Finally, the local surface temperature responses are decomposed as


$$\Delta T = -\frac{\Delta LW^\uparrow_{TOA,clr,\varepsilon}}{D} - \frac{\Delta LW^\uparrow_{TOA,cld,\varepsilon}}{D} + \frac{\Delta SW^\downarrow_{TOA,clr}}{D} + \frac{\Delta SW^\downarrow_{TOA,cld}}{D} + \frac{\Delta SW^\downarrow_{TOA,Albedo}}{D} - \frac{\Delta F^\downarrow_{SURF}}{D} \quad (11)$$
$$+ \frac{\Delta\left(C^\rightarrow - \frac{\delta E}{\delta t}\right)}{D}$$
$$= -\Delta LW_{clr,\varepsilon} - \Delta LW_{cld,\varepsilon} + \Delta SW_{clr} + \Delta SW_{cld} + \Delta SW_{Albedo} + \Delta SURF + \Delta CONV.$$

In the above equation, the temperature responses related to the first five components build up from a sum of the instant radiative forcing (if any), rapid adjustments associated with the component, and a temperature dependent feedback which adjusts its magnitude as the surface temperature changes, normalized by $D$ (the Planck feedback). Therefore, temperature responses related to these terms are functions of a constant term (forcing and adjustments) and a time dependent term (the impact of feedback due to surface temperature changes). For example, the LW flux response to a change in clear-sky longwave emissivity is $\Delta LW^\uparrow_{TOA,clr,\varepsilon} \approx F^\uparrow_{TOA,LW,clr,\varepsilon} - \lambda_{LR+LWWV}\Delta T$ in a linearized a forcing-feedback framework, where $F^\uparrow_{TOA,LW,clr}$ is the


longwave component of the effective radiative forcing and $\lambda_{LR+LWWV}$ is the combined longwave lapse-rate and water vapor feedback (e.g. Crook and Forster, 2011). Similarly, $\Delta LW^{\uparrow}_{TOA,cld,\varepsilon} \approx F^{\uparrow}_{TOA,LW,cld,\varepsilon} + \lambda_{LW\_cld}\Delta T$, $\Delta SW^{\downarrow}_{TOA,clr} \approx F^{\downarrow}_{TOA,SW,clr} + \lambda_{SWWV}\Delta T$, $\Delta SW^{\downarrow}_{TOA,cld} \approx F^{\downarrow}_{TOA,SW,cld} + \lambda_{SW,cld}\Delta T$, and $\Delta SW^{\downarrow}_{TOA,Albedo} \approx F^{\downarrow}_{TOA,SW,Albedo} + \lambda_{SW,Albedo}\Delta T$. $\Delta F^{\downarrow}_{SURF}$ and

$\Delta\left(C^{\rightarrow} - \frac{\delta E}{\delta t}\right)$ do not have direct counterparts in the linear forcing-feedback analysis, and they have been incorporated as part of the energy budget in regional forcing-feedback analysis in various ways in the literature (Crook et al., 2011; Feldl & Roe, 2013; Lu & Cai, 2009).

## 2.2 Decomposition of the standard deviation in surface temperature response


Decomposing the temperature responses $\Delta T_i$ also allows us to to decompose their contributions $CSD_i$ to the model-to-model standard deviations $\sigma_{\Delta T}$ of the total temperature responses by,

$$CSD_i = \frac{cov(\Delta T_i, \Delta T)}{\sigma_{\Delta T}} = r_i \sigma_{\Delta T_i}, \qquad (12)$$

where $cov(\Delta T_i, \Delta T)$ is the model-to-model covariance between $i$:th time-averaged local temperature response component and

the total local temperature response of a model experiment, $r_i$ and $\sigma_{\Delta T_i}$ are their model-to-model correlation and the standard deviation, respectively, and $\sigma_{\Delta T}$ is the standard deviation of time-averaged temperature responses in different models. $CSD_i$'s sum up to the inter-model standard deviations of the temperature responses,

$$\sum_i CSD_i = \sigma_{\Delta T}. \qquad (13)$$

## 2.3 Models and Simulations

We use climate model data from (PDRMIP) (Myhre et al., 2017). In PDRMIP, several independent climate models were used
to simulate various idealized climate perturbations. The models used in this study are listed in Table 1. According to Knutti (2013) all these models belong to different model families, and hence are independent from each other. Our study uses data from experiments of instant doubling of $CO_2$ concentrations (co2x2), tripling of $CH_4$ concentrations (ch4x3), five folding sulfate emission (sulx5) and ten folding black carbon emissions (bcx10) (see Table 2). Perturbations were relative to the





baseline which had the present day (models except HadGEM2-ES) or the pre-industrial (HadGEM2-ES) levels of anthropogenic forcing agents.

**Table 1. PDRMIP models used in this study, ocean and aerosol configuration of the model and which aerosol-cloud interactions are included.**

| Model | Ocean Setup | Aerosol setup | Interactive so4/bc | Key refenreces |
|---|---|---|---|---|
| **CanESM2** | Coupled | Emissions | yes/no | Arora et al., 2011 |
| **NCAR-CESM1-CAM4** | Slab ocean | Fixed concentrations | no/no | Gent et al., 2011 Neale et al., 2010 |
| **GISS-E2-R** | Coupled | Fixed concentration | no/no | Schmidt et al., 2014 |
| **HadGEM2-ES** | Coupled | Emissions | yes/no | Collins et al., 2011 Martin et al., 2011 |
| **NorESM1** | Coupled | Fixed concentrations | yes/yes | Bentsen et al., 2013; Iversen et al., 2013; Kirkevåg et al., 2013 |
| **MIROC-SPRINTARS** | Coupled | HTAP2 Emissions | yes/yes | Watanabe et al., 2010; Takemura et al., 2005; Takemura et al., 2009 |

**Table 2: Description of  PDRMIP experiments**





| Experiment name | Description |
| --- | --- |
| **baseline** | Anthropogenic forcing agents are at present day levels or at preindustrial levels |
| **co2x2** | Instantaneous doubling of the $CO_2$ concentration relative to the base case |
| **ch4x3** | Instantaneous tripling of the $CH_4$ concentration relative to the base case |
| **sulx5** | Five-folding the sulfate concentration or emissions relative to the base case |
| **bcx10** | Ten-folding black carbon concentration or emissions relative to the base case |

All simulations consisted of 100-year baseline and perturbed runs, and the last 50 years of these runs are used for the temperature response analysis carried out here. The PRDMIP experiments also included additional fixed sea-surface temperature runs, which we use for the calculation of the effective radiative forcing ($ERF_{fsst}$) associated with each climate perturbation. We also calculated the effective radiative forcing by regressing the top-of-atmosphere radiative imbalance against surface temperature change ($ERF_{reg}$) by using the full 100-year timeseries of experiments, as further discussed below. Aerosol emissions were either defined explicitly or by multiplying pre-defined concentrations derived from AeroCom Phase II (Myhre et al., 2012) (see Table 1). Only NorESM1 and MIROC-SPRINTARS include the aerosol indirect effect (the Twomey effect) for black carbon; however, the semi-indirect effect is included in all models. The aerosol cloud effects from sulfate are included





in all models except NCAR-ESM1-CAM4 and GISS-E2-R. We include the six PDRMIP models which had reported all necessary fields for this analysis.

**2.4 Global TOA radiative forcing and surface temperature responses of analyzed experiments**

In this paper, we focus on decomposed local and global temperature responses normalized by the global effective radiative forcing ($ERF_{fsst}$) obtained from fixed-sea-surface-temperature experiments for each modelled perturbation. The normalization by $ERF_{fsst}$ enables us to compare the temperature responses of different modelled perturbations with each other on a level ground, as $ERF_{fsst}$ varies in sign and magnitude between different perturbations. Also, particularly in aerosol and methane experiments the modelled $ERF_{fsst}$ values vary between different models, likely due to differences in model aerosol setups and

baseline methane concentrations.

Figure 2 shows the calculated effective radiative forcings and the global mean temperature responses (the difference in perturbed climate for the years 50-100 and the corresponding years from the base case) in the analyzed PDRMIP experiments. The effective radiative forcing is calculated from both fixed-sea-surface-temperature simulations ($ERF_{fsst}$, no land-warming

corrections included) and by regressing the top-of-atmosphere radiative imbalance with respect to surface temperature change by using the full 100-year timeseries of experiments ($ERF_{reg}$, Gregory et al., 2004).



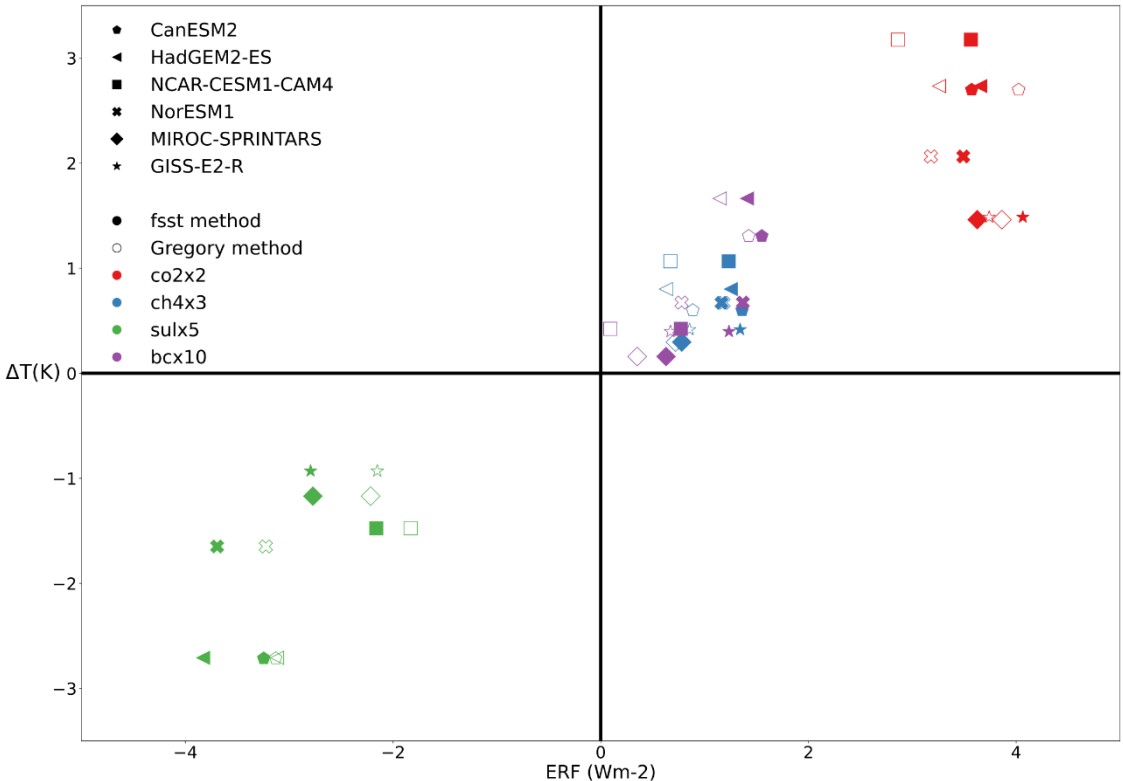

**Figure 2: The global average temperature responses for each experiment and each model (y-axis) averaged over years 50-100 after the sudden introduction of climate perturbations. The calculated ERFs for each experiment and model are shown on the x-axis, with non-filled marks indicating the $ERF_{reg}$ obtained using the Gregory method, and the filled markers indicating $ERF_{fsst}$ obtained from fixed-sea-surface-temperature simulations.**

One of the models (NCAR-CESM1-CAM) was ran using a slab ocean configuration, while the rest of the models contained fully interactive ocean configurations. Since the equilibrium is reached in a few decades with slab ocean configurations while with fully interactive ocean configuration it takes centuries, the perturbed experiments with models besides NCAR-CESM1-CAM are still in a transient state. As a multi-model mean over the years 50-100 of the perturbed runs, the doubling of $CO_2$ concentration (red marks) leads to a 2.27 K rise in global surface temperatures with a model-to-model standard deviation (std) of 0.65 K. The corresponding values are for tripling of $CH_4$ (blue marks) 0.64K (std 0.25K), five folding sulfate aerosols (green marks) –1.77 K (std 0.70K) and ten folding black carbon (purple) aerosols 0.77K (std 0.54 K). The exact numbers for each





forcer and model are shown in the supplementary (Tables S1-S4), and the estimated equilibrium temperature responses for each of the experiments is shown in Table S5.

The multi-model-mean $ERF_{fsst}$ values for co2x2, ch4x3, sulx5 and bcx10 experiments are, respectively, 3.66 Wm$^{-2}$ (std 0.19 Wm$^{-2}$), 1.19 Wm$^{-2}$ (std 0.19 Wm$^{-2}$), -3.08 Wm$^{-2}$ (std 0.58 Wm$^{-2}$), and 1.16 Wm$^{-2}$ (std 0.34 Wm$^{-2}$). When the effective forcings

are calculated from regressions using the full 100-year timeseries, the corresponding $ERF_{reg}$ values are 3.49 Wm$^{-2}$, (std 0.41 Wm$^{-2}$), 0.82 Wm$^{-2}$ (std 0.19 Wm$^{-2}$), -2.61 Wm$^{-2}$ (std 0.56 Wm$^{-2}$), and 0.74 Wm$^{-2}$ (std 0.45 Wm$^{-2}$). Tang (2019) has carried out complete analysis of $ERF_{fsst}$ and $ERF_{reg}$ for the PDRMIP data, and our values are consistent with the values presented there. We also refer the readers to Tang (2019) for the $ERF_{fsst}$ values obtained with the land warming correction accounted for, and for $ERF_{reg}$ calculated from the first 30 years of perturbed experiments.


Figure 2 show that only weak relationship between the model-to-model values in $ERF_{fsst}$ (or $ERF_{reg}$) and the model-to-model spread in temperature response can be seen for co2x2 and ch4x3 experiments, while some relationship exists for the 5xsulf and 10xbc experiments. Correlations between the models' temperature response and their $ERF_{fsst}$ are for co2x2 and ch4x3, -0.52 and 0.43 while with sulx5 and bcx10 correlations are 0.61 and 0.78. As also visible in Fig. 1 for individual models, the

application of the regression method for the full 100-year timeseries of experiments provides consistently lower values for ERF compared to values obtained from fixed sea surface temperature calculations. Overall, $ERF_{fsst}$ appears to be a more suitable choice for the surface temperature response normalization of different experiments due to very small values of $ERF_{reg}$ associated with some bcx10 experiments.

**3 Results**

In the following sections, we present decomposed effective temperature responses for each analyzed experiment and model-to-model spread of these decompositions. The effective surface temperature responses and their decompositions are calculated for each atmospheric column separately from the average differences in perturbed climates for the years 50-100 after a sudden

perturbation and the corresponding years from the baseline simulations without perturbations. The local temperature responses are normalized by the globally averaged $ERF_{fsst}$ for each experiment (hence the term effective). Scaling with $ERF_{fsst}$ allows a simpler comparison of responses between different forcing agents, but it also changes the sign of responses in case of sulx5 experiments. This is because in contrast to the other three forcing agents, the radiative forcing is negative for increasing sulfate concentrations


The local temperature responses related to longwave and shortwave TOA components build up from a combination of the local instant top-of-atmosphere radiative forcing and rapid adjustments associated with each term, and a temperature dependent



feedback which adjusts its magnitude as the surface temperature changes, as described in the end of Section 2.1. Therefore, temperature responses related to these components are functions of a constant term - forcing (if any) and rapid adjustments -
and a time dependent term - the impact of feedback as surface temperature changes.

The temperature response decomposition applied here relies on a local conservation of energy in each atmospheric column, and hence the sums of individual temperature response components generate the local total surface temperature responses with high accuracy. Below, Section 3.1 presents the globally averaged results. Section 3.2 then presents the regional distributions
of the decomposed surface temperature responses and their zonal averages. Section 3.3 presents the regional and latitudinal distributions of the model-to-model standard deviations of the effective temperature components, and the contributions of each of the decomposed surface temperature response components to the total standard deviations of the responses.

**3.1 Decomposed global effective surface temperature responses for different forcers**

Figure 3 shows the globally averaged effective surface temperature responses and their decomposed components for each model and perturbation experiment, calculated by using the temperature decomposition method described in Section 2.1. The components of the effective surface temperature responses describe the combined global contributions of the TOA forcing (in case of clear-sky $\Delta LW_{clr}$, $\Delta SW_{clr}$ components and $\Delta LW_{cld}$ and $\Delta SW_{cld}$ cloud components) and the effects of rapid adjustments
and feedbacks associated with each component. Of the components not associated with the TOA radiative forcing, $\Delta SW_{Albedo}$ is directly related to the response due to surface albedo feedback, $\Delta SURF$ is a measure of the surface energy flux imbalance on global surface temperatures due to oceanic heat uptake in the models, and $\Delta CONV$ describes the impact of horizontal surface energy transport and change in atmospheric heat uptake. The models which have a fully coupled ocean have not fully reached equilibrium, and therefore there is still some heat flux from the atmosphere to the ocean. Thus, the effective
temperature response from this heat flux is always negative. Globally, $\Delta CONV$ averages effectively to zero in each experiment since the energy transport only redistributes regional surface temperature effects, and the change in atmospheric heat uptake is negligible on annual or longer timescales (Räisänen, 2017).





**Figure 3. The global mean effective temperature response and its decomposition, calculated as the difference between means over the last 50 years of the perturbed and the baseline experiments. (a) The effective temperature response (absolute temperature response divided by ERF$_{fsst}$) for the six models shown with different symbols, and four different radiative forcers shown with different colors. (b) The decomposition of the effective temperature response to different energetic components. Individual panels in (b) describe (from the left) the contributions to total effective temperature response due to the change in longwave clear-sky emissivity ($\Delta LW_{clr,\epsilon}$), change in TOA shortwave radiation ($\Delta SW_{clr}$), change in longwave cloud emissivity ($\Delta LW_{cld,\epsilon}$), net ocean surface heat flux ($\Delta SURF$), and change in atmospheric energy transport ($\Delta CONV$).**

The total effective temperature responses (temperature response divided by the ERF$_{sst}$) for co2x2, ch4x3, sulx5, and bcx10 experiments are 0.63 KW$^{-1}$ m$^2$ (std 0.19), 0.54 KW$^{-1}$m$^2$ (std 0.18), 0.57 KW$^{-1}$ m$^2$ (std 0.18) and 0.61 KW$^{-1}$ m$^2$ (std 0.32), respectively. Hence, the mean value and the model-to-model spread in total effective temperature response is similar for




different forcers, as shown in previous PDRMIP studies (Richardson et al., 2018; Samset et al., 2018; Stjern et al., 2017). The change in TOA longwave clear-sky emissivity is the key driver of the effective temperature response for the greenhouse gas experiments (co2x2 and ch4x3), with the multi-model-mean effective surface temperature responses to $\Delta LW_{clr}$ matching nearly
exactly the overall responses (0.60 KW$^{-1}$ m$^2$ ± 0.10 and 0.53 KW$^{-1}$ m$^2$ ± 0.18 respectively). $\Delta LW_{clr}$ results from the change in clear-sky planetary emissivity, i.e. from the combination of the longwave clear-sky radiative forcing and its adjustments, and the change in the thermal structure of the atmosphere and water vapor concentrations which evolve with the surface temperature response (lapse rate and water vapor feedbacks). For the aerosol experiments (sulx5 and bcx10) the effective temperature response associated with $\Delta LW_{clr}$ is only a small contribution to the total temperature response. This is because for
aerosols the instantaneous radiative forcing associated with the longwave clear-sky radiation is small.

The differences in effective temperature responses associated with the $\Delta SW_{clr}$ between the greenhouse gas and aerosol forcers can be understood via a similar narrative as in case of $\Delta LW_{clr}$ responses. The total effective temperature response for aerosol experiments (sulx5 and bcx10) is largely dominated by the temperature response to $\Delta SW_{clr}$ , since much of instantaneous
aerosol radiative forcing takes place via this channel. For the greenhouse gas experiments the temperature response to $\Delta SW_{clr}$ originates from the shortwave water vapor feedback and direct GHG's shortwave forcing (Etminan et al., 2016), and its model-mean contribution to total effective temperature response is comparable to that from the albedo response (~10%).

The multi-model-mean effective temperature responses related to $\Delta LW_{cld}$ and $\Delta SW_{cld}$ are close to zero for all experiments
besides for bcx10, for which the cloud temperature responses modestly oppose the total effective temperature response. For the sulx5 experiment, the model-mean $\Delta SW_{cld}$ is near zero and its spread is high between the models. A significant part of the spread is related to the lack of cloud-aerosol interaction in NCAR-CESM1-CAM4 and GISS-E2-R. In these models, $\Delta SW_{cld}$ reduces the sulfate-induced global mean cooling, whereas it amplifies the cooling in the other models.

The global effective temperature response from the changes in surface albedo is similar across each experiment. The mean effective temperature response due to albedo change varies from ~10% (co2x2, chx3 and bcx10) to 14% (sulx5) of the total effective temperature response. In the aerosol experiments the aerosol setup has a significant effect on the temperature contribution of the surface albedo change. Emission-driven models tend to produce a higher albedo effective temperature response than the concentration-driven models.


**3.2 Origins of regional temperature responses for different climate forcers**

The model-mean spatial distributions of effective temperature responses and their decomposed components are shown in Fig. 4. The zonal means of different components are shown in Fig. 5, where we have summed up the contributions of surface and
atmospheric energy transport components ($\Delta SURF$ and $\Delta CONV$) due to their strong tendency to balance each other regionally.





Furthermore, the total response due to clouds ($\Delta LW_{cld}$ and $\Delta SW_{cld}$) is shown in Figure 5 together with individual cloud components. Again, we remind the reader that scaling all results with $ERF_{fsst}$ changes the sign of responses in sulx5 experiments.

**Figure 4. The multi-model mean effective temperature response (row 1) for four different climate forcers, i.e. carbon dioxide (column 1), methane (column 2), sulfate (column 3) and black carbon (column 4), and its decomposition to different energy balance terms. (Long- and shortwave clear-sky ($\Delta LW_{clr}$, $\Delta SW_{clr}$), clouds, surface energy exchange ($\Delta SURF$) and horizontal energy transport ($\Delta CONV$)). Dotted areas show regions where only 3-4 out of the 6 models agree on the sign of the response.**

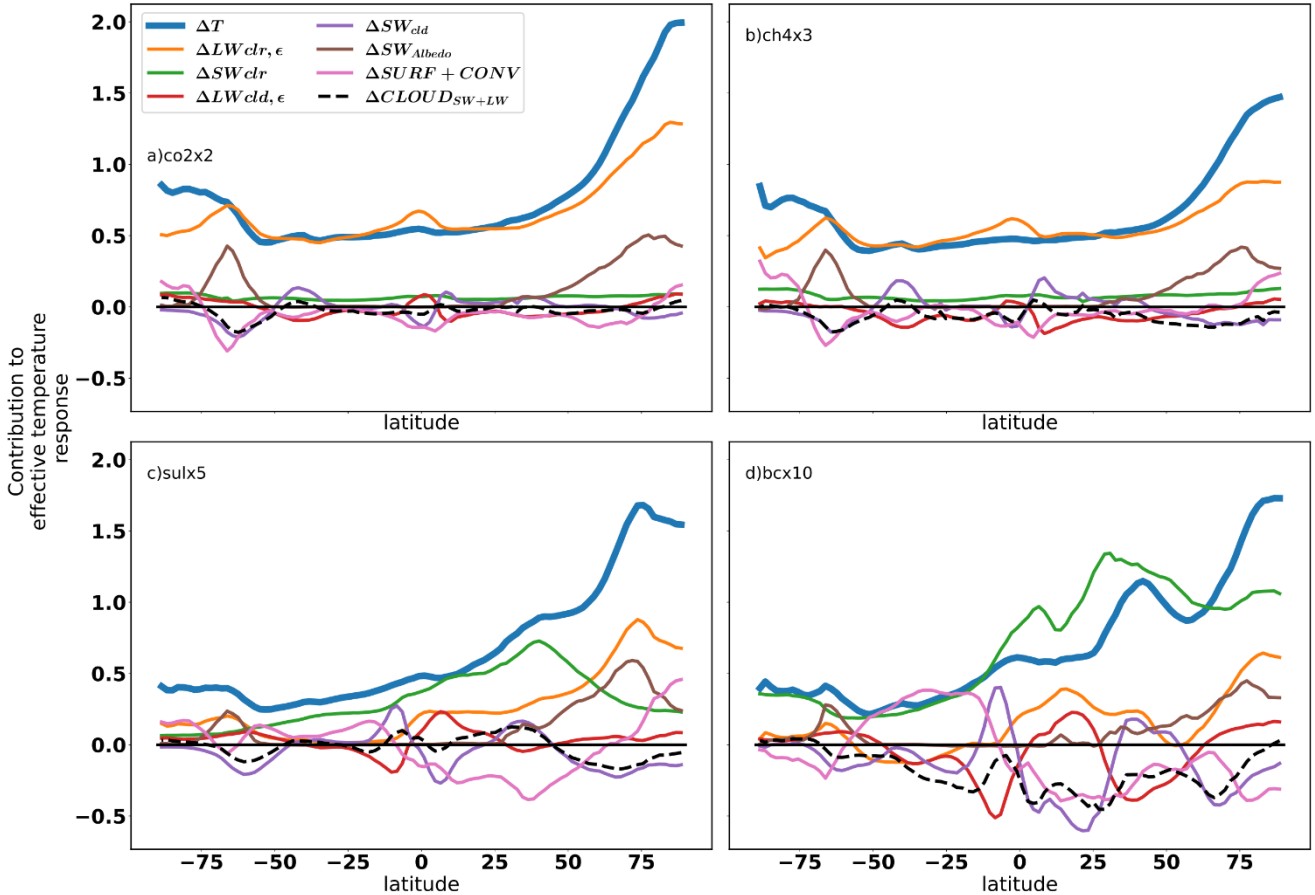

**Figure 5: Zonal average multi-model mean effective temperature response (thick blue lines) and its decomposition into different energetic terms (thin coloured lines) for different climate forcers. Panel (a) shows co2x2, panel (b) ch4x3, panel (c) sulx5 and panel (d) bcx10 experiment.**

The spatial distribution of the total effective temperature response is largely similar for each forcer, although the total response to aerosols is stronger over the continental Northern midlatitudes, compared to total responses to greenhouse gases, and weaker over the Southern hemisphere oceans. Regionally, local maximum effective temperature responses are found in the Barents Sea for all forcers, with maximum values of 2.38, 2.04, 2.96 and 2.53 $KW^{-1} m^2$ for co2x2, ch4x3, sulx5 and bcx10, respectively. Stejrn et al. (2019) showed that the largest local temperature responses in PRDMIP experiments are in regions with the largest sea ice changes. Differences between forcers can be seen for example over the Antarctic region where both greenhouse gas experiments (ch2x2, ch4x3) produce Antarctic amplification which is not seen in the aerosol experiments. The effective





temperature responses in the bcx10 experiment show higher contrasts between land and sea regions than in the other experiments.

For greenhouse gases, the regional effective temperature responses are mostly associated with the response to $\Delta LW_{clr}$. However, with all forcers the spatial distribution of the $\Delta LW_{clr}$ contribution resembles the overall effective temperature response. The spatial correlation coefficients between the effective total and $\Delta LW_{clr}$ -induced temperature responses for the co2x2, ch4x3, sulx5 and bcx10 experiments are 0.90, 0.81, 0,94 and 0.74, respectively. The difference between the greenhouse gas and aerosol experiments is that for greenhouse gases the $\Delta LW_{clr}$ response includes the combined effects of forcing, its
rapid adjustments and lapse rate and water vapor feedbacks, while for aerosols the response results only from the rapid adjustments and lapse rate and water vapor feedbacks. For the co2x2 and ch4x3 experiments, the total effective temperature response and $\Delta LW_{clr}$ temperature response differ most in the equatorial Pacific Ocean, where the $\Delta LW_{clr}$ response exceeds the total response. The high $\Delta LW_{clr}$ temperature response in this region is compensated by contributions from $\Delta SW_{cld}$, $\Delta LW_{cld}$ and atmospheric energy transport $\Delta CONV$ (see Fig, 4).


For aerosols, most of the effective temperature response is due to $\Delta SW_{clr}$ . Besides the modest water vapor contributions to $\Delta SW_{clr}$ , this response is directly related to excess scattering and absorption of solar radiation (direct aerosol radiative forcing) due to changes in aerosol concentrations, as was shown in Merikanto et al. (2021). Most of the sulfate emissions originate from Asia, Europe and North America, while most of the black carbon emissions originate from Asia, Europe and North
America and African, South American and boreal wildfires (Myhre et al., 2012), This makes the forcing in both cases stronger in the Northern than in the Southern hemisphere. The $\Delta SW_{clr}$ temperature response to these emissions can be clearly seen both for the bcx10 and sulx5 experiments (see Fig. 5). For the bcx5 experiments, the local effective temperature response due to $\Delta SW_{clr}$ exceeds the total effective temperature response from the tropics to the Northern midlatitudes (Fig. 5). These local excess warming responses by $\Delta SW_{clr}$ in bcx5 experiments are counteracted by temperature responses to changes in atmospheric
energy transport and clouds. In case of aerosols, $\Delta LW_{clr}$ contributes to the effective temperature response mainly in the Northern hemisphere continents and Arctic sea-ice regions. In the bcx10 experiments, $\Delta LW_{clr}$ induces a clear negative contribution over ocean regions related to changes in the vertical temperature distribution of the atmosphere (see Fig. S2).

There is significant variation in the regional effective temperature contributions due to clouds between regions and forcing
agents. In the greenhouse gas experiments (co2x2 and ch4x3) the regional effective temperature responses due to $\Delta LW_{cld}$ and $\Delta SW_{cld}$ tend to cancel each other, except over the Southern Ocean where the total cloud contribution is dominated by a negative $\Delta SW_{cld}$ (see Fig. 5). This relates to a strong increase in cloud cover in the same regions (see Figures S4 and S5).

With aerosols, the net effect of clouds is more complicated. Both the sulx5 and bcx10 experiments show a similar negative
effective temperature response due to $\Delta SW_{cld}$ over the Southern Ocean as the greenhouse gases. However, Northern



hemispheric cloud responses are larger in magnitude for aerosols than for greenhouse gases, and for the bcx10 experiment in particular. Throughout the latitudes bcx10 causes a strong net cloud cooling, except for the polar regions where the net cloud responses are small. The sign of the regional $\Delta SW_{cld}$ effective temperature response in the bcx10 experiments depends strongly on the region. There is a negative temperature response due to $\Delta SW_{cld}$ in Asia and Africa, but positive over the Amazon

region. This is related to the cloud cover change, as black carbon increases the cloud cover over Asia and Africa, but decreases it over the Amazon (see Fig, S5) due to the semi-direct aerosol effect of black carbon. Contrary to bcx10, sulx5 shows a mild positive contribution from clouds over Northern hemispheric midlatitudes and a mild cooling response in the Arctic regions. However, the strength of the $\Delta SW_{cld}$ responses in the sulx5 experiments depends on the inclusion or lack of aerosol-cloud indirect effect in the models.


The effective temperature response to surface albedo change originates from the change in sea-ice and snow cover and is always positive. Changes in surface albedo have a modest effect on the global effective temperature response with all forcers (0.07,0.06,0.08,0.06 $KW^{-1} m^2$ for the co2x2, ch4x3, sulx5, bcx10 experiments, respectively). However, in some regions, the effective temperature response to albedo change exceeds 1 $KW^{-1} m^2$ for all forcers. Over the Arctic, the regions of local

maximum values are the same where the overall effective temperature responses are highest, highlighting the role of sea-ice changes causing locally high temperature responses. The local maximum values of the effective temperature response to albedo change also match with the regions with a positive temperature response to ocean heat exchange ($\Delta SURF$). The change in surface albedo also enhances the temperature response over the Southern Ocean, but there the temperature response to oceanic heat exchange is slightly negative. However, over the Southern Ocean the temperature response signal to surface albedo change

is mainly visible in the co2x2 and ch4x3 experiments, and appears to be driven by the longwave clear-sky forcing and feedbacks, and ocean heat transport.

Over the oceans, $\Delta SURF$ has a large impact on the effective temperature response, with the co2x2, ch4x3 and sulx5 experiments all showing negative effective temperature contributions due to $\Delta SURF$ south of Greenland and positive

contributions over the Barents Sea. With black carbon a robust negative signal over the northern North Atlantic is missing, however, but similarly to other forcers there is a robust positive signal over the Barents Sea. As earlier found for increased $CO_2$ by Räisänen (2017), the effects of oceanic heat transport and storage ($\Delta SURF$) and atmospheric heat transport ($\Delta CONV$) strongly oppose each other over the oceans in the co2x2, ch4x3 and sulx5 experiments. In the sulx5 and bcx10 experiments, $\Delta CONV$ also significantly compensates for the surface temperature effects due to changes in $\Delta SW_{clr}$, which reflects changes

in the direct aerosol forcing.

**3.3 Model-to-model spread in regional effective temperature responses for different forcers**





Similarly, to the effective temperature response itself, also its model-to-model spread (standard deviation) can be decomposed
to components that sum up to the total spread in the effective surface temperature response (Sect. 2.2). Figure 6 shows the
decomposed model-to-model standard deviations of the total effective temperature responses (first row) for each perturbation
experiment, and the decomposed contributions of each component to the spread in total responses. The latitudinal distributions
of the different components are shown in Fig. 7.

**Figure 6. The model-to-model standard deviation of the effective temperature response to different climate perturbations (row 1)
and its decomposed different energetic components (rows 2-8). Each column shows results for four different climate forcers, i.e.
carbon dioxide (column 1), methane (column 2), sulfate (column 3) and black carbon (column 4). The global mean values are shown
at the bottom-right corner of each panel.**

**Figure 7. Zonal mean of the total standard deviation of the effective temperature response (thick blue line), and the contributions of the different energy balance terms to it (thin lines, see the legend in (a)). Panel (a) shows co2x2, panel (b) ch4x3, panel (c) sulx5 and panel (d) bcx10 experiment.**

The globally averaged magnitude of the model-to-model spread is similar between co2x2, ch4x3 and sulx5 experiments (0.19, 0.18, and 0.18 KW$^{-1}$m$^2$, respectively). Black carbon induces a much larger variability between the models (0.32 KW$^{-1}$m$^2$). The spatial structure of the model-to-model spread resembles the spatial structure of the effective temperature response. Furthermore, the spread in the temperature response amplifies towards polar regions, but the polar amplification of the spread is even stronger than the amplification of the responses. Indeed, the majority of the model spread comes from the sea ice regions in the high latitudes, but the location of the maximum model-to-model spread varies between forcers. With co2x2 and ch4x3, the regions with highest model spread are in the Arctic Ocean region north of Siberia (1.10 KW$^{-1}$m$^2$ with co2x2 and 1.30 KW$^{-1}$m$^2$ with ch4x3) and in the Labrador Sea (0.90 KW$^{-1}$m$^2$ with co2x2, and 1.60 KW$^{-1}$m$^2$ with ch4x3). With sulfate, the





spread is the largest in the ocean region between Iceland and Svalbard (2.47 KW$^{-1}$m$^2$), and for black carbon east of Svalbard
(2.23 KW$^{-1}$m$^2$). In the co2x2 and ch4x3 experiments, the strongest component in model-to-model spread is $\Delta LW_{clr}$ (see Fig. 7), but the partial contributions of other components are also significant. The amplification of the spread in the effective temperature response towards high latitudes (Fig. 7) is strongly related to additional spread arising from differences in the surface albedo response ($\Delta SW_{Albedo}$), reflecting differences in sea ice and snow cover responses. The total contributions of cloud responses to the model spread are significant over Southern and Northern midlatitudes and to a lesser extent over the equatorial region. In the Southern Ocean sea-ice regions, the model spread originates from differences in the $\Delta LW_{clr}$ and $\Delta SW_{Albedo}$ responses in the models, as well as from differences in the oceanic heat exchange ($\Delta SURF$) compensated by differences in the atmospheric heat transport ($\Delta CONV$) (see Fig. 6). Between 30-45 °S, the model spread due to the combined effect of clouds ($\Delta SW_{cld}$+ $\Delta LW_{cld}$) is also evident. However, in the co2x2 and ch4x3 experiments the $\Delta SW_{cld}$ and $\Delta LW_{cld}$ terms often oppose each other, thus making the combined contribution of clouds to the total model spreads small in these experiments.

In the aerosol experiments (sulx5 and bcx10) the build-up of the model-to-model spread is more complicated than for the greenhouse gas experiments, despite similarities in the latitudinal distribution of the total spread of the effective temperature response. The contributions of $\Delta SW_{clr}$ (the pathway of aerosol direct radiative forcing) and the combined cloud response ($\Delta SW_{cld}$+ $\Delta LW_{cld}$) to the total model spread are much more significant in the aerosol experiments than in the greenhouse gas experiments. In the aerosol experiments, $\Delta SW_{clr}$ adds to the model spread over the Southern Ocean in both the sulx5 and bcx10 experiments, suppresses the model spread over midlatitude and equatorial oceans in sulx5 and over Southern hemisphere and equatorial continents in bcx10, and adds model spread over Northern hemispheric continents (bcx5) and over the Arctic Ocean (both sulx5 and bcx10). Clouds have a large impact on the regional model-to-model spread in the aerosol experiments, and dominate the zonal means of the model spread in the sulx5 experiments outside of the polar regions. Much of the model spread related to the combined cloud contributions ($\Delta SW_{cld}$+ $\Delta LW_{cld}$) results from differences in the aerosol setups in the models. With aerosol experiments (sulx5 and bcx10) most of the cloud-induced model-to-model spread is related to emissions sources, and is highly affected by which aerosol-cloud effects are included in the models. Also for aerosols, the contributions to model spread due to $\Delta SW_{cld}$ and $\Delta LW_{cld}$ often oppose each other, but the stronger model spread related to the $\Delta SW_{cld}$ response dominates the overall cloud contribution in midlatitudes, while model spread due to $\Delta LW_{cld}$ dominates the model spread over the equatorial regions.

On the other hand, in the aerosol experiments (sulx5 and bcx10) the atmospheric heat transport ($\Delta CONV$) tends to compensate the regional differences in model responses more efficiently than in the greenhouse gas experiments. In addition, the total heat transport ($\Delta SURF$ + $\Delta SURF$) reduces the zonally averaged model spread (see Fig. 7), particularly in the case of the bcx10 experiments. Similarly to the greenhouse gas experiments, $\Delta LW_{clr}$ is still a major driver of model-to-model spread also in the aerosol experiments, and particularly in the high latitude regions (Fig. 6). Compared to the greenhouse gas experiments, the





aerosol experiments exhibit much greater contributions from $\Delta SW_{Albedo}$ to the overall model-to-model spread in the Arctic region. In the co2x2 and ch4x3 experiments, the maximum contributions to the model spread in the Arctic due to $\Delta SW_{Albedo}$ are 0.23 and 0.29 KW$^{-1}$ m$^2$, respectively, while for sulx5 and bcx10 the corresponding values are 0.32 and 0.41 KW$^{-1}$m$^2$, respectively. In the Southern hemisphere high latitudes aerosols and greenhouse gases have a similar structure in model-to-model variability, however the aerosol experiments do not show a clear signal from $\Delta SW_{cld}$ in the Southern Ocean.

Previously, the model-to-model spread in global climate sensitivity (equilibrium response to doubled $CO_2$ concentration) has been largely attributed to differences in cloud feedback strength (Colman, 2003; Zelinka et al., 2020; Zhao et al., 2016). Our results point to somewhat divergent conclusions. Similarly to Hu et al. (2020), our results point to the water vapor feedback as the main mechanism leading to model spread. If the model spread is only attributed using feedback analysis, model differences in the forcing and adjustments may counteract some of the differences. However, it should be noted that our results are based on only six different models, and hence might be biased. For example, Zelinka et al. (2020) show that the contribution of clouds to the equilibrium climate sensitivity response exhibits notably large variation from approximately -0.2K to 3K for CMIP6 models and -0.18K to 2.6K for CMIP5. In our results $\Delta SW_{cld}$ and $\Delta LW_{cld}$ largely cancel each other out in the co2x2 experiments, leading to a smaller combined cloud contribution to the model spread and contributes only 12% (see table S1) to the global model spread. For comparison, for a sample of 16 CMIP5 models with a transient response to doubling of $CO_2$, Räisänen (2017) found the clear-sky LW response to be the largest contributor to the model spread in 34% of the global area, whereas the combined cloud response had this position in 29% of the world.

## 4 Discussion and Conclusions

In this work, we have conducted an energy balance decomposition of the near-surface temperature response resulting from doubling $CO_2$, tripling $CH_4$, five folding sulfate concentrations, and ten folding black carbon concentrations for six independent climate models. The regional temperature response was then decomposed to contributions from different energy balance terms, namely changes in LW and SW clear-sky and cloud radiative fluxes (SW and LW), the net surface energy flux (SURF), and horizontal energy transport (CONV). All forcers produce a similar global response per unit ERF (0.63, 0.54, 0.57 and 0.61 KW$^{-1}$ m$^2$ for increasing $CO_2$, $CH_4$, sulfates and black carbon, respectively). The majority of the globally averaged temperature change for doubling the $CO_2$ and tripling the $CH_4$ concentration, originates from changes in clear-sky planetary emissivity (0.60 and 0.53 KW$^{-1}$ m$^2$ respectively), i.e. from the combination of the longwave clear-sky radiative forcing with the change in the thermal structure of the atmosphere and water vapor concentrations. In the aerosol experiments (sulfate and black carbon) the key driver of surface temperature response is the change in the clear-sky shortwave radiative flux (0.36 and 0.71 KW$^{-1}$ m$^2$ respectively) related to excess scattering and absorption of solar radiation (direct aerosol radiative forcing) due to changes in aerosol concentrations. We note that if only models including sulfate aerosol-cloud interactions are included, the SW cloud terms are in the same magnitude ($\Delta SW_{clr}$=0.25 and $\Delta SW_{cld}$=0.12 KW$^{-1}$ m$^2$). See below for a more detailed discussion on different aerosol setups. The overall temperature response to the forcers is largest in high latitudes, where the response is driven





by changes in LW clear-sky fluxes and changes in surface albedo, except for black carbon for which the majority of the total
       response comes from the clear-sky SW term in the high latitudes.

       The temperature decomposition method provides a tool for understanding regional and global temperature changes. However,
       the original method is somewhat simplistic in its treatment of LW cloud processes (Räisänen, 2017). In Merikanto et al. (2021)

we implemented a radiative kernel correction to make the LW treatment of clouds more realistic. However, despite this
       correction we still have a negative effective LW temperature response from clouds when the $CO_2$ concentration is doubled,
       whereas literature suggests a positive LW cloud feedback (Tomassini et al., 2013; Vial et al., 2013). This is due to neglecting
       the positive masking effect of increasing $CO_2$ on the LW cloud forcing, since the corresponding effects can not be calculated
       for the other forcing agents with existing kernels. The applied radiative kernels for the temperature and $H_2O$ have a relatively

minor  effect on global averages of the LW cloud and clear-sky terms (see Fig, S6) . Similarly to previous studies (Smith et
       al., 2018), we found that the relative importance of the kernel correction does not depend on the radiative kernel used.

       In our study, clouds play a minor role in the global mean temperature response, as  the LW cloud and SW cloud terms tend to
       cancel each other out. However, regionally the temperature response originating from the clouds is a significant contributor.

For all forcers, the temperature response in the Antarctic sea-ice region and in the Southern Ocean is dampened by clouds.
       With BC clouds dampen the regional temperature response in Asia, North America, Africa, and Europe, and enhance the
       warming in Amazon. In contrast to clouds, with all forcing agents surface albedo changes enhance the temperature responses
       in high latitudes. For greenhouse gases, the mild polar amplification in south is associated with a negative contribution from
       the ocean heat exchange over the Southern Ocean, negative total cloud contribution and a mild LW clear-sky component.


       We also decompose the model-to-model spread to the contributions of energy balance terms. The model-to-model spread is
       the largest in the same regions as the average temperature response, i.e. at high latitudes, where the spread is driven by
       differences in the lapse-rate and water vapor feedbacks ($\Delta LW_{clr}$) and differences in surface albedo ($\Delta SW_{Albedo}$) changes. For
       the aerosol-induced temperature responses, also differences in the direct aerosol forcing ($\Delta SW_{clr}$) generate a significant

contribution to the model spread, especially for black carbon. This partly arises from different aerosol configuration between
       models.

       The aerosol configuration is important in the generation of the effective temperature response and its model-to-model spread.
       In the aerosol experiments, part of the model-to-model spread originates from the difference between aerosol setups, with the

emission-driven models generating a higher effective temperature response than the concentration-driven models. For sulx5,
       the concentration-driven models' mean effective temperature response is 0.49 KW $^{-1}$ m$^2$ while for the emission-driven models
       it is 0.66 KW $^{-1}$ m$^2$. The corresponding numbers for the bcx10 experiments are 0.45 KW $^{-1}$ m$^2$ and 0.76 KW $^{-1}$ m$^2$. For the sulx5
       experiments, the sign and the regional distribution of $\Delta SW_{cld}$ is strongly related to the aerosol setup; however, it should be



noted that two out of the three concentration-driven models do not include aerosol-cloud interactions. In the bcx10 experiment,
the aerosol setup modifies the $\Delta SW_{clr}$ and $\Delta LW_{clr}$ temperature responses (see S1, showing the decomposed temperature
responses separately for the concentration and emission-driven models for the sulx5 and bcx10 experiments). The aerosol
configuration also plays a crucial role in the SW-albedo response. For both the sulx5 and bcx10 aerosol experiments, only the
emission-driven models show a significant temperature contribution from the $\Delta SW_{Albedo}$ term (0.1 and 0.08 KW $^{-1}$ m$^2$
respectively) whereas the corresponding mean values for the concentration driven models are only 0.06 and 0.03 KW $^{-1}$ m$^2$.

We have demonstrated that the mechanisms behind model uncertainty vary between different regions and forcing agents.
Understanding the atmosphere's dynamical response to different forcers is key to understand future climate changes at the
regional level. This is especially important in the case of aerosols, which are predicted to decline in the near future due to
climate change and air pollution mitigation actions.

**Data and code availability**

Data and scripts used for data analysis can be obtained by contacting corresponding author

**Author contributions**

The manuscript was written by KN and JM, with contributions from all authors. KN and JM performed the analysis with help
of JR. BS provided the PDRMIP data.

**Competing interests:**

The authors declare that they have no conflict of interest


**Acknowledgements**

We gratefully acknowledge the efforts of the PDRMIP community, and the modelers who have kindly made their simulation
results publicly available. Storage and availability of PDRMIP data was provided by UNINETT Sigma2 - the National
Infrastructure for High Performance Computing and Data Storage in Norway. BHS acknowledges funding by the Research
Council of Norway, project nr. 244141 (NetBC). JR acknowledges funding by Academy of Finland Flagship funding (grant.
nr 337549). KN acknowledges funding by Academy of Finland grant nr. 340791.



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
