# Peer review of "Understanding the surface temperature response and its uncertainty to CO2, CH4, black carbon and sulfate"

_Atmospheric Chemistry and Physics, 2021_

## Author Response (AR1)

Response to Reviewer 1

We would like to thank Reviewer 1 for the constructive comments and suggestions. All changes are marked with red color to revised manuscript

- 47: see also Gillett et al. 2021: https://www.nature.com/articles/s41558-020-00965-9
  - citatation added. Since the cooling effect of aerosols in the Gillett et al. was estimated to be lower than that in the original references, we also included this estimate and changed the sentence to "Modern day anthropogenic aerosols cool the global surface temperatures between 0.1-1.1 °C."
- 56: Furthermore, the differences in regional distributions of aerosol surface temperature responses are not dominated by the aerosol description in modern climate models (Nordling et al., 2019). I'm a bit confused by this sentence.
  - The sentence is now changed to: Furthermore,the differences in aerosol surface temperature response between modern climate models are not dominated by the aerosol description (Nordling et al., 2019).
- fig1. last panel should be (f),
  - This is now fixed
- 117: How should the reader interpret \Delta \T and \Delta \epsilon_{eff}? Are they local changes?
  - Yes, they are local changes. The paragraph below Equation 2 clarifies that all changes are local.
- 186: maybe "largely" independent. There's still some possible cross-pollination of code bases and research teams.
  - Changed according to the suggestion.
- 233-234 and further on: stylistic, but using $\pm$ rather than std looks cleaner.
  - Changed according to the suggestion.
- 247: here 5xsulf and 10xbc, previously sulx5 and bcx10
  - changed to sulx5 and bcx10
- 267: instant -> instantaneous
  - changed
- 268-270: suggest slightly revising the grammar of this sentence as the multiple dashes are confusing.
  - This has been done. The dashes have been removed.
- Figure 5: Add units somewhere in either the caption or y-label axis. I think it's K/(W/m2).
  - Units have been added The same was done for fig 7.
- 357-358: Barents Sea - will have to take your word for it as not too obvious from the resolution of figure 4!
  - Color scale has been changed, extremes are now more visible. This is also more visible in the LW clear-sky term.
- 382: bcx10 or sulx5?
  - this typo is fixed
- 485: $\Delta$SURF + $\Delta$SURF
  - typo fixed

We would like to thank Review 2 for valuable comments and suggestions.We have revised the manuscript according to the suggestions. All changes are marked with red color to revised manuscript

- It would be useful if panel a) of figure 1 could be labelled with all the terms used in section 2.1. Or if this panel is too small, a separate schematic of showing all the terms would be useful.
  - The panel is too small for this. Furthermore, all terms used in section 2.1 are local changes, whereas panel a) shows an atmospheric column without perturbations. Hence the terms in section 2.1 are not present in panel a.
- Line 105: This could clarify at the start that all these fluxes are net, I first interpreted the arrows as meaning upwards and downwards components until I realised they were net.
  - This is done as suggested
- Line 105: You use the term "cloud radiative fluxes" here, rather than "cloudy-sky radiative fluxes", I presume this is deliberate and part of the APRP decomposition. If so, this should be more explicit at this point.
  - This line text is changed to namely to changes in local longwave fluxes associated with changes in clear-sky and cloud emissivity ( LWclr and LWcld , respectively, with the arrow indicating the vector direction towards space) at TOA, changes in shortwave fluxes due to changes in clear-sky absorption and reflection as well changes in cloudiness and cloud radiative properties ( SWclr , and SWcld),
- Line 120: This is defined (eqn 2) as the change in OLR associated with the change in the local effective emissivity of the planet (not atmosphere). It might also be worth clarifying that changes in the effective emissivity of the planet and changes in the atmospheric emissivity have opposite signs.
  - Emissivity of the atmosphere is here changed to planetary emissivity
- Equation 3: It might make more sense for the arrow on C to be leftward rather than rightward since it is inward flux.
  - Changed as suggested.
- Line 144: Some brief explanation of how the CS and CRE decomposition is done would be helpful. Is CRE just the cloudy-sky component?
  - Here CRE is just the cloudy-sky component. A short text is added to line 150 where we say that these terms are calculated using Equation 2
- Equation 11: I found the terminology on the second line of this equation confusing since DeltaLW and DeltaSW have units of K rather than W/m2. Couldn't you use DeltaT with different subscripts?
  - We found that using DeltaT's with suffixes in the text became too messy. To Clarify that the terms in the last line of Eq. 11 are in units of temperature, we added a text to the end of the paragraph: "In the last line of Eq. (11) the terms without TOA suffix, that is the radiative components divided by $D$, are in units

of temperature. Hereafter, we will use these terms as shorthand notations when discussing the various temperature responses in the text. "

- Line 169: I think there should be a minus sign rather than a plus before the lambda_LW_cld DeltaT term? Or at least it should have the opposite sign to all the downward terms.
  - Fixed
- Line 204: I think it is better to say the semi-indirect effect is "inherent" in all models since it is not something than can be explicitly included or excluded. It would be better to use "meteorological adjustments" rather than semi-direct.
  - Done as suggested
- Section 2.4: Why do you not include land-warming corrections? Tang et al. and Richardson et al. show they are important for CO2.
  - This is included as suggested in Figure S5.
- Line 246: The relationship might be stronger if you remove the land component from the ERFfsst.
  - This is partly true, for co2 and and black carbon the correlations are higher, however, for sulfate and methane correlations are smaller than for any other method. A new figure (S5) is added to the supplementary material to show ERF with land warming correction. Due to the small number of models, the sensitivity of the relationship to different methods of estimating ERF is difficult to assess..
- Line 309: This seems to imply that all the feedback processes are manifesting themselves in the LW and in the clear-sky which is slightly surprising. It might be worth signalling that you will discuss why this differs with Zelinka in the discussion section.
  - Line 323 includes now  The large model to model spread compared to other terms is discussed more in section 3.3. In section 4 we discuss why these differs from example values presented by Zelinkta et al. (2020). The somewhat large spread in LW clearsky terms in probably due to small number of models
- Line 314: The feedbacks however should appear in DeltaLWclr. Zelinka suggest the WV+LR feedback is around half the Planck feedback (DeltaT in your figure). Does this imply the feedbacks are different for aerosols and not simply a function of surface temperature change?
  - This is not implying that feedback are different for greenhouse gases and aerosols. With aerosols, the forcing is more on SW side than on LW, whereas DeltaLWclr for greenhouse gases includes a substantial contribution from forcing. The temperature response depends both on forcing and the feedback, as discussed at the end of section 2.1.
- Line 325: Presumably this the offsetting cloud adjustment found in earlier PDRMIP papers (Stjern et al.).?
  - This is true, Stjern et al. (2017) shows that rapid adjustments for BC are dominated by the cloud response, and the rapid adjustments dampen the overall temperature response. Line 338 includes now: . With the bcx10 the net cloud effect is cooling across different latitudes, despite variations between $\Delta LW_{cld}$ and $\Delta SW_{cld}$. The increase of low level clouds over the Arctic regions and reductions of clouds in upper troposphere (see fig. S4) due to BC forcing is typical cloud response and these dominates the rapid adjustments and leads dampening of the surface response (Stjern et al. 2017).

- Line 329: For models with an indirect effect I would expect DeltaSWcld to be as large (or larger) than the direct effect. Does this imply that there is a negative SWcld feedback that adds (negatively) to the indirect effect?
    - Yes it does. The changes in cloud cover dominate the overall response.
- Figure 5: needs units
    - Units added.
- Line 387: Can you explain more how the temperature distribution changes in bcx10 and why that means a negative DeltaLWclr? Is it greater LR than WV feedback, or is the initial adjustment?
    - The following line is added: The top-heavy warming in bcx10 experiment results from fast adjustments as shown in Smith et.al 2018.
- Line 404: Could you separate the DeltaSWcld for the models with and without indirect effects?
    - SWcld terms for each model are now added to the supplementary
- Figure 7: Needs unit
    - Units added.
- Figure S2: This would be easier to interpret if it were divided by ERF
    - Done as suggested. Figures S2 to S4 includes now both absolute values and values normalized by ERF.

---

## Author Response (AR2)

We would like to thank Editor for addressing small comments which adds clarity to this paper

There are a few minor point I ask you to address prior o publication of your revised manuscript. I am referring to line numbers of version 2 of the manuscript.

-- Lines 55/56: you re-ordered the sentences that reviewer 1 identified as confusing. However, this does not make the sentence any clearer. Please try to reword - maybe the term 'aersosol description' needs further explanation

Line 55-56 is now referring models athropogeni aerosol desrcitpions

-- Line 239: (-± 0.25K) likely should be ($\pm$\,0.25\,K) - skipping the minus before \pm ?"

-Typo fixed

-- Line 108: indices cld do not match with indices clr and cld in the response to reviewer 2 - please doublecheck if all indices are correct here

-Typo fixed

-- Line 323: 'these differs' should be 'this differs' or 'these differ'. It is unclear what 'these' refers to - please clarify

-Line 323 replace these by model-to-model spread

-- Line 400: the added statement on the bcx10 experiment does not fully answer the reviewer's question - please extend

-This is extended "where BC absorbs in coming SW-radiations leading warming of the upper atmosphere and increase low level clouds. Future more, over oceans there is more moisture available for low level cloud formations leading cooling of the surface"